# Defining Benchmarks for Continual Few-Shot Learning

## Abstract

In recent years there has been substantial progress in few-shot learning, where a model is trained on a small labeled dataset related to a specific task, and in continual learning, where a model has to retain knowledge acquired on a sequence of datasets. Both of these fields are different abstractions of the same real world scenario, where a learner has to adapt to limited information from different changing sources and be able to generalize in and from each of them. Combining these two paradigms, where a model is trained on several sequential few-shot tasks, and then tested on a validation set stemming from all those tasks, helps by explicitly defining the competing requirements for both efficient integration and continuity. In this paper we propose such a setting, naming it *Continual Few-Shot Learning (CFSL)*. We first define a theoretical framework for CFSL, then we propose a range of flexible benchmarks to unify the evaluation criteria. As part of the benchmark, we introduce a compact variant of ImageNet, called *SlimageNet64*, which retains all original 1000 classes but only contains 200 instances of each one (a total of 200K data-points) downscaled to $64 \times 64$ pixels. We provide baselines for the proposed benchmarks using a number of popular few-shot and continual learning methods, exposing previously unknown strengths and weaknesses of those algorithms. The dataloader and dataset will be released with an open-source license.

## 1 Introduction

Two capabilities vital for an intelligent agent with finite memory are *few-shot learning*, the ability to learn from a handful of data-points, and *continual learning*, the ability to sequentially learn new tasks without forgetting previous ones. Taken individually these two areas have recently seen dramatic improvements mainly due to the introduction of proper benchmark tasks and datasets used to systematically compare different methods (Chen et al., 2019; Lesort et al., 2019a; Parisi et al., 2019). For the set-to-set few-shot setting (Vinyals et al., 2016) such benchmarks include Omniglot (Lake et al., 2015), CUB-200 (Welinder et al., 2010), Mini-ImageNet (Vinyals et al., 2016) and Tiered-ImageNet (Ren et al., 2018b). For the single-incremental-task continual setting (Maltoni & Lomonaco, 2019) and the multi-task continual setting (Zenke et al., 2017; Lopez-Paz & Ranzato, 2017) the benchmarks include permuted/rotated-MNIST (Zenke et al., 2017; Goodfellow et al., 2013), CIFAR10/100 (Krizhevsky et al., 2009), and CORe50 (Lomonaco & Maltoni, 2017). However, none of those benchmarks is particularly well suited for evaluating the hybrid setting of low-data sequential streams.

One of the main reasons behind the scarce consideration of the liaison between the two settings is that these problems have been often treated separately and handled by two distinct communities. Historically the research on continual learning has focused on the problem of avoiding the loss of previous knowledge when new tasks are presented to the learner, known as *catastrophic forgetting* (McCloskey & Cohen, 1989), without paying much attention to the low-data regime. On the other hand, the research on few-shot learning has mainly focused on achieving good generalization over new tasks, without caring about possible future knowledge gain or loss. Scarce attention has been given to few-shot learning in the more practical continual learning scenario.

In this paper we propose to bridge the gap between the two settings by injecting the sequential component of continual learning into the framework of few-shot learning, calling this new paradigm *Continual Few-Shot Learning (CFSL)*. CFSL can be useful to the research community as a frame-

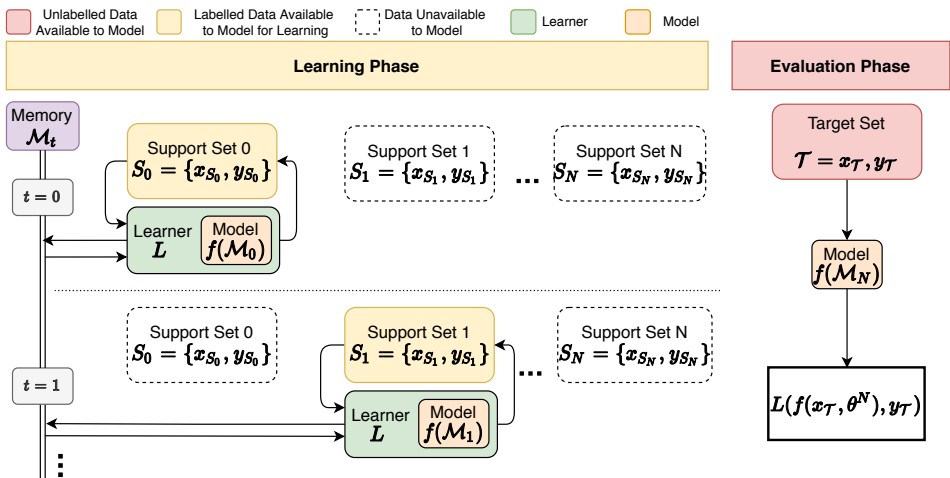

Figure 1: High level overview of the proposed benchmark. *Left block*: from the left, the learner acquires task-specific information from each set, one-by-one, without being allowed to view previous sets (memory constraint). The learner can store knowledge in a shared memory bank and use it in a classification model. On the rightmost side, future tasks are inaccessible to the learner. On the bottom, the same process is repeated on the second support set. Note that the first support set is now inaccessible. *Right block*: once the learner has viewed all support sets, it is given an evaluation set (target set) containing new examples of classes contained in the support-sets, and tasked with producing predictions for those samples. The evaluation procedure has access to the target set labels, and can establish a generalization measure for the model.

work for studying continual learning under memory constraints, and for testing meta-learning systems that are capable of continual learning. While we formally define the problem in Section 3, a high-level diagram is shown in Figure 1. Our *main contributions* can be summarized as follows:

1. We formalize a highly general and flexible continual few-shot learning setting, taking into account recent considerations expressed in the literature.

2. We propose a novel benchmark and a compact dataset (SlimageNet64), releasing them under an open source license.

3. We compare recent state-of-the-art methods on our benchmark, showing how CFSL is effective in highlighting the strengths and weaknesses of those methods.

## 1.1 MOTIVATION AND APPLICATIONS

Consider a user in a fast changing environment who must learn from the many scenarios that are encountered. There is the significant challenge of integrating common information from very few data points in each scenario in an online fashion. The small number of data points makes this distinct from normal continual learning setting: the very high uncertainty in each scenario due to the low data volume makes adaptation more challenging and makes the commonality between scenarios even more critical. The online nature of learning makes this distinct from few-shot learning, where integration from different scenarios must be learnt without access to earlier data. These two requirements for online learning without forgetting and efficient integration under uncertainty are competing: both require memory capacity of the learner. Because of these competing requirements, it is valuable to consider continual learning and few-shot learning together rather than in isolation.

For a concrete example, consider typical user interfaces such as those used in online stores. The size of data points collected from each user is rather small (few-shot) and is generally stored in a sequential buffer or priority queue (continual). Suppose an underlying learning model has been deployed to enhance the user experience by suggesting new products that are likely to be of interest. This model should be able to rapidly adapt to each user (task) by accessing the sequential buffer while learning on the fly. There are multiple variants to take into account. For instance, if the user is unknown or previous data is not accessible (e.g. under privacy policies) the model has to rapidly infer the user preferences from a single task. On the other hand, if the user profile is known the model should retain knowledge about previous interactions without the need of being retrained

from scratch. Another example arise from human-robot interaction, where most of the applications require a robot to learn online by interacting with human teachers. For instance, in a manipulation task the human can provide a few trajectories representing the first task, then the second, the third, etc. The amount of trajectories for each task is usually rather limited and the tasks are learned sequentially. The robot should retain the knowledge of all tasks encountered so far, possibly by avoiding expensive training procedures that would overload the on-board hardware.

Note that neither the few-shot nor the continual setting are appropriate to deal with the aforementioned examples, since the former does not consider the sequential component and the latter does not account for the limited data size. Our CFSL formulation instead, can handle all these examples and other collateral variations, as discussed more thoroughly in Section 3.

## 2 RELATED WORK

### 2.1 FEW-SHOT LEARNING

Progress in few-shot learning (FSL) was greatly accelerated after the introduction of the episodic few-shot training (Vinyals et al., 2016). This setting, for the first time, formalized few-shot learning as a well defined problem paving the way to the use of end-to-end differentiable algorithms that could be trained, tested, and compared. Among the first algorithms to be proposed there were meta-learned solutions, which here we group into three categories: metric-learning, optimization-based, hallucination (Chen et al., 2019). *Metric-learning* techniques are based on the idea of parameterizing embeddings via neural networks and then use distance metrics to match target points to support points in latent space (Vinyals et al., 2016; Edwards & Storkey, 2017; Snell et al., 2017). *Optimization-based* or *gradient-based* techniques are trained to perform a controlled optimization or parameter initialization to learn efficiently from a support set and generalize to a target set (Ravi & Larochelle, 2016; Li et al., 2017; Finn et al., 2017; Antoniou et al., 2019; Rusu et al., 2019; Antoniou & Storkey, 2019). *Hallucination* techniques utilize one or both the aforementioned methods in combination with a generative process to produce additional samples as a complement to the support set (Antoniou et al., 2017). There have been a number of methods that do not clearly fall in one of the previous categories (Santoro et al., 2017; Santurkar et al., 2018; Chen et al., 2019), including Bayesian approaches (Grant et al., 2018; Gordon et al., 2019; Patacchiola et al., 2019). For more detail, we refer the reader to the original work as well as a survey on few-shot learning (Chen et al., 2019).

### 2.2 CONTINUAL LEARNING

The problem of continual learning (CL), also called life-long learning, has been considered since the beginnings of artificial intelligence and it remains an open challenge in machine learning (Parisi et al., 2019). In standard offline supervised learning, algorithms can usually access any data point as many times as necessary during the training phase. In contrast, in CL data arrives sequentially and might only be ever seen once during the training process. Following the taxonomy of (Maltoni & Lomonaco, 2019), we group the continual learning methods into three classes: architectural, rehearsal, and regularization methods. Each category brings with it a different set of advantages and disadvantages under various resource constraints. *Architectural* approaches can be constrained on the amount of available RAM (Rusu et al., 2016; Mallya et al., 2018; Mallya & Lazebnik, 2018; Lesort et al., 2019a). Whereas, *rehearsal* strategies can become quickly bounded by the amount of available storage (Rebuffi et al., 2017; Lesort et al., 2018; 2019b). *Regularization* approaches can be free from resource constraints but incur in severe issues in the way they adapt model parameters (Kirkpatrick et al., 2017; Zenke et al., 2017; Lee, 2017; He & Jaeger, 2018; Mitchell et al., 2018). The mentioned strategies can often be intersected and combined to form even more powerful models (Rebuffi et al., 2017; Kemker et al., 2018; Maltoni & Lomonaco, 2019). Due to space constraints, we refer the reader to recent surveys on continual learning (Lesort et al., 2019c; Parisi et al., 2019).

### 2.3 JOINING META-LEARNING AND CONTINUAL LEARNING

Attempts to combine continual-learning with meta-learning produced a set of new research areas (Caccia et al., 2020). Continual Few-Shot Learning falls into *meta continual-learning* which can

also be thought of as 'learning to continually learn' (Finn et al., 2019; He et al., 2019; Harrison et al., 2019). In contrast, *continual meta-learning* refers to 'continually learning to learn' which attempts to make the process of meta-learning continuous as opposed to the standard meta-learning which is typically performed offline. Very recently Caccia et al. (2020) proposed a hybrid task, called Online faSt Adaptation and Knowledge Accumulation (OSAKA), linking continual-meta learning and meta continual-learning to study continual learning in the context of non-stationarity on shifting task distributions and unknown identities. Related to continual few-shot learning is the field of *incremental few-shot learning* (IFSL, (Gidaris & Komodakis, 2018; Ren et al., 2018a)). In contrast to standard few-shot learning and our work, in IFSL target set is composed of 'novel' classes (drawn from a never-seen-before dataset) as well as classes seen during the meta-learning phase (called 'base classes'), and it does not consider continual updates to the novel classes. These methods are significantly different in terms of training and testing procedures. For this reason, we will not analyze thie line of research any further.

**Inconsistencies in the evaluation protocol** In the literature, there are no established benchmarks that integrates few-shot and continual learning. Related tasks were introduced to prove the efficacy of a given system, making such tasks very restricted in terms of what methods they are applicable on, and how many aspects they could investigate. We found that tasks and datasets vary from paper to paper, making it challenging to know the actual performance of a given algorithm in comparison to others. For instance, the method proposed by Vuorio et al. (2018) has been tested exclusively on variants of MNIST. The method of Javed & White (2019) has been tested on Omniglot and incremental sine-waves. Spigler (2019) evaluated on MNIST, Le et al. (2019) on CIFAR100 and permuted MNIST, and Beaulieu et al. (2020) on Omniglot. It is evident how the problem of continual few-shot learning is not well defined, making it challenging to benchmark and compare the performance of algorithms.

## 3 CONTINUAL FEW-SHOT LEARNING

### 3.1 DEFINITION OF THE PROBLEM

A continual few-shot learning (CFSL) task is composed of a sequence $\mathcal{G}$ of small training sets (support sets) $\mathcal{G} = \{\mathcal{S}_n\}_{n=1}^{N_G}$, and a small evaluation set (target set) $\mathcal{T}$. Each support set $\mathcal{S} = \{(\mathbf{x}_n, y_n)\}_{n=1}^{N_S}$ is a set of input-label pairs just like in the standard few-shot learning setup (Chen et al., 2019). A target set $\mathcal{T} = \{(\mathbf{x}_n, y_n)\}_{n=1}^{N_T}$ is a set of input-label pairs containing previously unseen instances of classes stemming from $\mathcal{G}$. The objective of the learner is to perform well on the validation set $\mathcal{T}$ having only temporal access to the labeled data contained in the support $\mathcal{S}$.

The size of the support set $N_S$ is defined by the number of classes $N_C$ (way) and by the *number of samples per class K* (shot), such that if we have a 5-way/1-shot setup we end up with $N_S = N_C \times K = 5 \times 1 = 5$. The *Number of Support Sets Per Task (NSS)* parameter determines $N_G$, the cardinality of $\mathcal{G}$. A *Class-Change Interval (CCI)* parameter dictates how often the classes within the sequence $\mathcal{G}$ should change, expressed in numbers of support sets. This corresponds to assigning the elements in the support sets to a series of disjoint class sets $\bigcap_{i=1}^{I} \mathcal{C}_i = \{\emptyset\}$, where $I = \lceil NSS/CCI \rceil$ and $\mathcal{C}_i$ is a set of unique classes of size $N_C$. For example, if CCI=2 then we will draw support sets whose classes change every 2 samples. As a result, support sets $\mathcal{S}_1$ and $\mathcal{S}_2$ will contain different instances of the same class set $\mathcal{C}_1$, whereas $\mathcal{S}_3$ and $\mathcal{S}_4$ will contain different instances from the class set $\mathcal{C}_2$. The process of generating CFSL tasks is also described in Algorithm 1 (Appendix E).

A *learner* is a process which extracts task-specific information and distills it into a classification model. The model can be generically defined as a function $f(\mathbf{x}, \boldsymbol{\theta})$ parameterized by a vector of weights $\boldsymbol{\theta}$. At evaluation time the learner is tested through a loss function

$$\mathcal{L} = \Big( f(\mathbf{x}_{\mathcal{T}}, \boldsymbol{\theta}), y_{\mathcal{T}} \Big), \tag{1}$$

where $\mathbf{x}_{\mathcal{T}}$ and $y_{\mathcal{T}}$ are the input-output pairs belonging to the target set. Note that we intentionally provided a definition that is generic enough to fit into different methodologies and not restricted to the use of neural networks.

To remove the possibility of converting a continual learning task to a non-continual one, we introduce a restriction, which dictates that a support set $\mathcal{S}$ is sampled from $\mathcal{G}$ without replacement, and deleted once it has been used by the learner. The learner should never have access to more than one support set at a time, and should not be able to review a support set once it has moved to the next one. This restriction induces a strict sequentiality in the access of $\mathcal{G}$.

The setup we have described so far is very flexible, and it allows us to define a variety of different tasks and therefore to target different problems. In the following section we provide a description of those tasks and show that they are consistent with the continual learning literature.

## 3.2 TASK TYPES

In this section we define an empirical procedure under the form of specific task types. Our CFSL tasks fully cover the standard single-incremental task scenario (Lomonaco & Maltoni, 2017) while introducing an additional, super-class NI setting consistent with Domain-Incremental learning (van de Ven & Tolias, 2018). Specifically, task A, B, and D are equivalent to *New Instances (NI)*, *New Classes (NC)*, and *New Instances and Classes (NIC)* (Maltoni & Lomonaco, 2019), respectively. Task C captures the super-set NI setting where instances are sampled across super-classes, instead of being sampled from previously defined class categories. This task is most similar to Domain-Incremental learning (van de Ven & Tolias, 2018). A detailed comparison between our tasks and existing work is reported in Appendix D. Figure 2 showcases a high-level visual representation of the proposed tasks.

A  **New Samples:** In this task type, each support set within a given task are sampled from the same set of preselected classes. As a result, each support set will share the same classes and labels but will contain previously unseen samples of those classes. To achieve this, we can set CCI to be equal to (or higher than) the number of support sets in a given task (CCI $\geq$ NSS). For every support set we sample new instances from the same classes as seen in previous support sets of the same task. The standard supervised regime can be considered as a particular case of this task.

B  **New Classes:** In this task type, each support set has different set of classes from the other support sets within a given task (CCI = 1). Each class within each support set has a corresponding unique output unit in the model. Class-incremental learning is a particular case of this task.

C  **New Classes with Overwrite:**. This task is similar to task B in that each support set has different set of unique classes (CCI = 1). The difference is that each class in the overall task is grouped and assigned a new label by *overwriting* the true label. As a result the number of output units in the model is equivalent to the number of unique classes within a single support set $\widetilde{\mathcal{C}}$. Intuitively, $\widetilde{\mathcal{C}}$ could be the hierarchical categories of classes in $\mathcal{G}^y = \cup_{n=1}^{N_G} \mathcal{S}_n^y$, however, in our experiments we assign the hierarchical categories arbitrarily. This setting emulates situations where an agent is tasked with learning data-streams while being limited in storing knowledge in a preselected number of output labels.

D  **New Classes with New Samples:** Combines tasks A and B, where each support set contains different instances of the same set of classes for some predefined class change interval (1 <CCI<NSS). The CCI defines when the classes (and labels) change to a different and disjoint set. This setting emulates situations where an agent is tasked with both learning new class descriptors and updating such descriptors by observing new class instances. It sheds light on how agents can perform on a setting that mixes all previous settings into one.

## 3.3 METRICS

**Test Generalization Performance.** A proposed model should be evaluated on at least the test sets of Omniglot and SlimageNet, on all the tasks of interest. This is done by presenting the model with a number of previously unseen continual tasks sampled from these test sets, and then using the target set metrics as the task-level generalization metrics. To obtain a measure of generalization across the whole test set the model should be evaluated on a number of previously unseen and unique tasks. The mean and standard deviation of both accuracy and performance should be used as generalization measures to compare models.

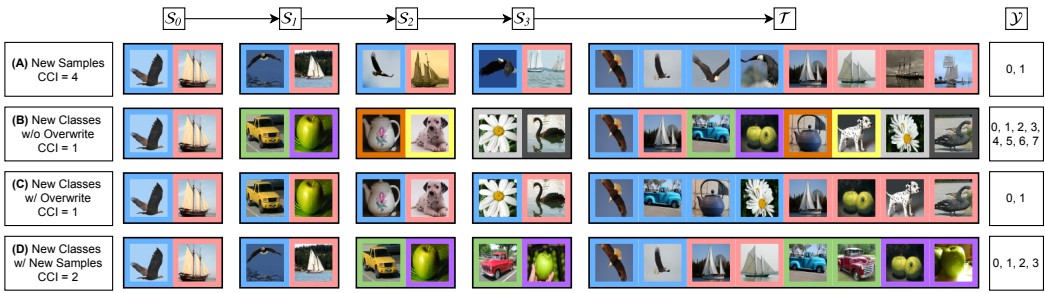

Figure 2: Visual representation of the four continual few-shot task types. Each row corresponds to a task with Number of Support Sets, NSS=4, and a defined Class-Change Cnterval (CCI). Given a sequence of support sets, $\mathcal{S}_n$, the aim is to correctly classify samples in the target set, $\mathcal{T}$. Colored frames correspond to the associated support set labels.

**Multiply-Addition operations (MACs)** Measures the computational expense of both the learner and the model operations during learning and inference time. In other words, it measures the memory footprint that the model itself needs to execute during a cycle of inference, or a meta-learning cycle.

**Across Task Memory (ATM).** Even though we have imposed a restriction on the access to $\mathcal{G}$, the learner is still authorized to store in a local *memory bank* $\mathcal{M}$ some representations of the inputs and/or output vectors (often implemented as embedding vectors or inner loop parameters)

$$\mathcal{M} = \{(\hat{\mathbf{x}}, \hat{y})_{\mathcal{S}_1}, ..., (\hat{\mathbf{x}}, \hat{y})_{\mathcal{S}_{N_G}}\},\tag{2}$$

where $\hat{\mathbf{x}}$ and $\hat{y}$ are representations of $\mathbf{x}$ and $y$ obtained after a given learner has processed $\mathbf{x}$ and $y$ and stored some of their useful components. Most learners will be compressing a given support set, but this is not strictly the case.

The potential compression rate is not directly correlated to the complexity of the model (e.g. number of parameters, FLOPs, etc). For instance, compression can be achieved by removing some of the dimensions of the input, or by using a lossless data compression algorithm, which may not require additional parameters or may have minimal impact on the execution time. In this regard, the concept of memory bank $\mathcal{M}$ helps to disambiguate model complexity from any additional memory allocated for compressed representations of inputs. We can use the cardinality of $\mathcal{M}$, indicated as $|\mathcal{M}|$, to quantify the learner efficiency. Given two learners with their corresponding models $f(\mathbf{x}, \boldsymbol{\theta}_1)$ and $f(\mathbf{x}, \boldsymbol{\theta}_2)$, and assuming that the size of $\boldsymbol{\theta}_1$ is equal to the size of $\boldsymbol{\theta}_2$ with $\mathcal{L}_1 = \mathcal{L}_2$, then the learner with smaller cardinality $|\mathcal{M}|$ must be preferred.

In order to compare performances across different tasks and datasets, we relate the size of the stored task-specific representations (in bytes) $\mathcal{M}^{\hat{x}}$ (e.g. embedding vectors in ProtoNets, and inner loop parameters for MAML) during task-specific information extraction to the size of vectors (in bytes) $\mathbf{x}$ contained in the episode $\mathcal{G}^x = \cup_{n=1}^{N_G} \mathcal{S}_n^x$. Recall that $\hat{\mathbf{x}}$ is a compressed version of $\mathbf{x}$ and therefore $F < H$, where $H$ and $F$ are the vectors of dimensionality of $\mathbf{x}$ and $\hat{\mathbf{x}}$, respectively. To reduce the notation burden we have only considered the inputs $\mathbf{x}$ and not the targets $y$, since $\mathbf{x}$ is significantly larger than $y$. Based on these considerations we define Across-Task Memory (ATM)

$$\text{ATM} = \frac{|\mathcal{M}^{\hat{x}}|}{|\mathcal{G}^x|},\tag{3}$$

where $\mathcal{M}^{\hat{x}}$ is the stored representation of a series of support sets and $\mathcal{G}^x$ is the size of the support sets. For each utilized floating point arithmetic unit we include a computation that takes into account the floating point precision level, as stored in tensor form. For example, if both $\mathcal{M}^{\hat{x}}$ and $\mathcal{G}^x$ use the same floating point standard then it is divided out, but if the representational form uses a lower precision than the actual data-points then it becomes compressive. From a practical standpoint (image classification), the ATM can be estimated relating the total number of bytes stored in the memory bank (ATM numerator) with the total number of bytes over all the images in the episode (ATM denominator). Given the definition above: ATM $< 1$ for learners with efficient memory, ATM $= 0$ for learners with no memory, and ATM $> 1$ for learners with inefficient memory. Note that the ATM is undefined for empty episodes $\mathcal{G} = \{\emptyset\}$. ATM is task/dataset agnostic and can be used to compare various models (or the same model) across different settings.

### 3.4 CHOICE OF DATASETS

The proposed CFSL benchmark has to be supplemented with appropriate datasets. The choice of these datasets should be considered decoupled from the benchmark itself. However, building some of the tasks requires datasets that meet specific desiderata: very high degree of diversity in terms of classes, high number of categories, fair (but not overabundant) number of samples per class, contained in size and memory footprint. The popular Omniglot (Lake et al., 2015) is a good first choice for a lower-difficulty dataset. Finding a higher complexity dataset (e.g. with RGB images) is arduous since existing datasets do not satisfy all the desiderata, being insufficient for robust and extensive benchmarking of CFSL algorithms (see detailed comparison in Appendix B).

**SlimageNet64.** To solve this issue, we propose a new variant of ImageNet64×64 (Chrabaszcz et al., 2017), named *SlimageNet64* (derived from Slim and ImageNet) consisting of 200 instances from each of the 1000 object categories of the ILSVRC-2012 dataset (Krizhevsky et al., 2012; Russakovsky et al., 2015), for a total of 200K RGB images with a resolution of $64 \times 64$ pixels. In Appendix B (Table 2) we report a detailed comparison of all the datasets available, showing how SlimageNet64 is an optimal choice over multiple criteria.

**SlimageNet64 vs Tiered-ImageNet.** The closest alternative to SlimageNet64 is Tiered-ImageNet (Ren et al., 2018b), a subset of ILSVRC-12 with a total of 608 classes. We have found three main issues with Tiered-ImageNet: (i) it is not provided as a stand-alone archive but aggregated via a generating script (different scripts have been released leading to an incompatibility between versions); (ii) it has just 608 classes, meaning a reduced diversity in the distribution space; (iii) it has a large memory footprint (29 GB size) that makes it impractical for training a large number of models in a reasonable time. On the other hand, SlimageNet64 contains more classes (1000) and it has a lower memory footprint (9 GB size) due to the smaller resolution of the images. This makes SlimageNet64 more compact since both dataset and model can be easily loaded on a single GPU.

## 4 EXPERIMENTS

For the purposes of establishing baselines in the CFSL tasks outlined in this paper we chose to use eight existing methods: (1) randomly initializing a convolutional neural network, and fine tuning on incoming tasks; (2) pretraining a convolutional neural network on all training set classes, then fine-tune on sequential tasks (Chen et al., 2019); (3) Elastic Weight Consolidation (EWC, Kirkpatrick et al. 2017) on a network which has been initialized and then finetuned (EWC-Init) as a baseline for CL methods; (4) EWC on a network which has been pretrained and finetuned (EWC-Pretrain); (5) Prototypical Networks (Snell et al., 2017) as baseline for metric-based FSL methods; (6) the Improved Model Agnostic Meta-Learning or MAML++ L (Low-End) model (Antoniou et al., 2019) as baseline for optimization based FSL methods; (7) MAML++ H (High-End) model (Antoniou & Storkey, 2019) (dense-net backbone, squeeze excite attention, mid-tier baseline); and (8) the Self-Critique and Adapt model (SCA) (Antoniou & Storkey, 2019), as a state-of-the-art algorithm for FSL (high-tier baseline). Additional details are reported in Appendix A. Table 1 and Figure 3 show the results obtained in the experimental section.

**Results: accuracy.** *(i) Omniglot.* The results on Omniglot in the New Classes without Overwrite Setting (B) MAML++ Low-End is inferior to ProtoNets, whilst in the New Classes with Overwrite Settings (C) this result is reversed. We infer that embedding-based methods are better at retaining information from previously seen classes, assuming that each new class remains distinct. However, when overwriting is enabled this trend is overturned because ProtoNet prototypes are shared by a number of super-classes containing categories that are harder to semantically disentangle. Gradient based methods such as MAML++ dominate in this case, since they can update their weights towards new tasks, and therefore achieve a better disentanglement of super-classes. SCA and High-End MAML++ (which utilize both embeddings and gradient-based optimization) produce the best performance across all settings. In the New Samples Setting (A), gradient based methods tend to outperform embedding-based methods while hybrid methods produce the best results. Furthermore, in the New Classes and Samples Setting (D), embedding-based methods outperform gradient-based methods, whilst hybrid methods continue to produce the best performing models. Regarding CL methods, we notice that the performance of EWC is close to the baseline conditions (Init & Tune, Pretrain & Tune), showing that EWC is not particularly effective in the CFSL setting. *(ii) Slim-ageNet.* Results on SlimageNet show that ProtoNets consistently outperform Low-End MAML++,

Table 1: Accuracy and standard deviation (percentage) on the test set for the proposed benchmarks. Best in bold.

| Task Type | B | C | A | D | B | C | A | D | B | C | A |
|---|---|---|---|---|---|---|---|---|---|---|---|
| NSS | 3 | 3 | 3 | 4 | 5 | 5 | 5 | 8 | 10 | 10 | 10 |
| CCI | 1 | 1 | 3 | 2 | 1 | 5 | 5 | 2 | 1 | 1 | 10 |
| Overwrite | False | True | True | False | False | True | True | False | False | True | True |
| **Omniglot** | | | | | | | | | | | |
| Init & Tune | $10.87_{\pm0.01}$ | $27.51_{\pm0.01}$ | $44.76_{\pm0.01}$ | $8.74_{\pm0.01}$ | $6.15_{\pm0.01}$ | $24.52_{\pm0.01}$ | $45.30_{\pm0.01}$ | $3.93_{\pm0.01}$ | $3.12_{\pm0.01}$ | $22.16_{\pm0.01}$ | $45.64_{\pm0.01}$ |
| Pre. & Tune | $9.97_{\pm0.14}$ | $26.75_{\pm0.27}$ | $32.44_{\pm0.29}$ | $7.91_{\pm0.15}$ | $6.02_{\pm0.02}$ | $24.51_{\pm0.06}$ | $31.89_{\pm1.10}$ | $3.86_{\pm0.06}$ | $3.13_{\pm0.03}$ | $22.30_{\pm0.06}$ | $33.17_{\pm0.39}$ |
| EWC-Init | $10.75_{\pm0.00}$ | $27.76_{\pm0.00}$ | $42.55_{\pm0.00}$ | $8.72_{\pm0.00}$ | $6.23_{\pm0.00}$ | $24.49_{\pm0.00}$ | $42.18_{\pm0.00}$ | $3.91_{\pm0.00}$ | $3.02_{\pm0.00}$ | $22.22_{\pm0.01}$ | $41.82_{\pm0.00}$ |
| EWC-Pre. | $9.55_{\pm0.00}$ | $26.23_{\pm0.00}$ | $30.27_{\pm0.00}$ | $7.73_{\pm0.00}$ | $5.93_{\pm0.00}$ | $24.20_{\pm0.00}$ | $30.15_{\pm0.00}$ | $3.71_{\pm0.00}$ | $3.23_{\pm0.02}$ | $22.10_{\pm0.00}$ | $30.14_{\pm0.00}$ |
| ProtoNets | $95.30_{\pm0.12}$ | $45.44_{\pm0.19}$ | $98.73_{\pm0.02}$ | $48.98_{\pm0.03}$ | $91.52_{\pm0.20}$ | $35.10_{\pm0.09}$ | $98.73_{\pm0.12}$ | $48.44_{\pm0.03}$ | $83.72_{\pm0.19}$ | $27.39_{\pm0.17}$ | $98.65_{\pm0.14}$ |
| MAML++L | $38.18_{\pm0.14}$ | $46.12_{\pm0.15}$ | $99.38_{\pm0.07}$ | $28.87_{\pm0.07}$ | $22.69_{\pm0.07}$ | $35.76_{\pm0.14}$ | $99.41_{\pm0.04}$ | $14.29_{\pm0.05}$ | $11.30_{\pm0.02}$ | $27.82_{\pm0.03}$ | $99.44_{\pm0.01}$ |
| MAML++H | $96.14_{\pm0.02}$ | $96.77_{\pm0.08}$ | $99.73_{\pm0.04}$ | $49.44_{\pm0.02}$ | $92.70_{\pm0.03}$ | $93.47_{\pm0.05}$ | $99.80_{\pm0.01}$ | $49.00_{\pm0.04}$ | $85.56_{\pm0.10}$ | $86.38_{\pm0.14}$ | $99.86_{\pm0.01}$ |
| SCA | $\mathbf{96.84_{\pm0.04}}$ | $\mathbf{97.38_{\pm0.02}}$ | $\mathbf{99.82_{\pm0.01}}$ | $\mathbf{49.71_{\pm0.01}}$ | $\mathbf{93.81_{\pm0.02}}$ | $\mathbf{94.08_{\pm0.45}}$ | $\mathbf{99.88_{\pm0.03}}$ | $\mathbf{49.51_{\pm0.01}}$ | $\mathbf{86.07_{\pm0.03}}$ | $\mathbf{87.29_{\pm0.19}}$ | $\mathbf{99.88_{\pm0.01}}$ |
| **SlimageNet64** | | | | | | | | | | | |
| Init & Tune | $8.4_{\pm0.01}$ | $21.3_{\pm0.01}$ | $24.4_{\pm0.01}$ | $6.1_{\pm0.01}$ | $4.5_{\pm0.01}$ | $20.8_{\pm0.01}$ | $24.7_{\pm0.01}$ | $3.0_{\pm0.01}$ | $2.4_{\pm0.01}$ | $20.5_{\pm0.01}$ | $24.9_{\pm0.01}$ |
| Pre. & Tune | $8.7_{\pm0.03}$ | $21.9_{\pm0.11}$ | $24.2_{\pm0.17}$ | $6.4_{\pm0.01}$ | $4.9_{\pm0.02}$ | $21.2_{\pm0.05}$ | $24.5_{\pm23}$ | $3.3_{\pm0.03}$ | $2.7_{\pm0.03}$ | $20.7_{\pm0.10}$ | $24.4_{\pm0.20}$ |
| EWC-Init | $8.51_{\pm0.00}$ | $21.66_{\pm0.00}$ | $24.81_{\pm0.00}$ | $6.11_{\pm0.00}$ | $4.60_{\pm0.00}$ | $20.98_{\pm0.00}$ | $24.87_{\pm0.00}$ | $2.93_{\pm0.00}$ | $2.38_{\pm0.00}$ | $20.39_{\pm0.00}$ | $24.67_{\pm0.00}$ |
| EWC-Pre. | $8.83_{\pm0.09}$ | $22.21_{\pm0.04}$ | $23.47_{\pm0.09}$ | $6.42_{\pm0.08}$ | $5.18_{\pm0.01}$ | $21.17_{\pm0.06}$ | $23.44_{\pm0.35}$ | $3.26_{\pm0.02}$ | $2.66_{\pm0.01}$ | $20.74_{\pm0.04}$ | $24.40_{\pm0.42}$ |
| ProtoNets | $24.1_{\pm0.05}$ | $25.9_{\pm0.23}$ | $43.1_{\pm0.24}$ | $15.1_{\pm0.03}$ | $18.2_{\pm0.14}$ | $22.7_{\pm0.09}$ | $43.3_{\pm0.03}$ | $10.4_{\pm0.12}$ | $12.3_{\pm0.09}$ | $21.0_{\pm0.06}$ | $43.7_{\pm0.15}$ |
| MAML++L | $13.6_{\pm0.04}$ | $25.5_{\pm0.23}$ | $42.7_{\pm0.10}$ | $10.2_{\pm0.11}$ | $7.9_{\pm0.13}$ | $22.6_{\pm0.03}$ | $43.0_{\pm0.12}$ | $5.0_{\pm0.08}$ | $3.6_{\pm0.14}$ | $20.8_{\pm0.09}$ | $43.0_{\pm0.42}$ |
| MAML++H | $27.2_{\pm0.25}$ | $33.8_{\pm0.16}$ | $61.2_{\pm0.36}$ | $16.8_{\pm0.18}$ | $21.0_{\pm0.21}$ | $\mathbf{30.4_{\pm0.51}}$ | $68.6_{\pm0.47}$ | $12.3_{\pm0.11}$ | $14.4_{\pm0.12}$ | $25.7_{\pm0.10}$ | $75.6_{\pm0.10}$ |
| SCA | $\mathbf{27.9_{\pm0.16}}$ | $\mathbf{34.0_{\pm0.23}}$ | $\mathbf{65.3_{\pm0.15}}$ | $\mathbf{17.3_{\pm0.07}}$ | $\mathbf{22.0_{\pm0.18}}$ | $30.1_{\pm0.36}$ | $\mathbf{72.0_{\pm0.36}}$ | $\mathbf{12.7_{\pm0.08}}$ | $\mathbf{14.6_{\pm0.07}}$ | $\mathbf{26.3_{\pm0.13}}$ | $\mathbf{77.4_{\pm0.06}}$ |

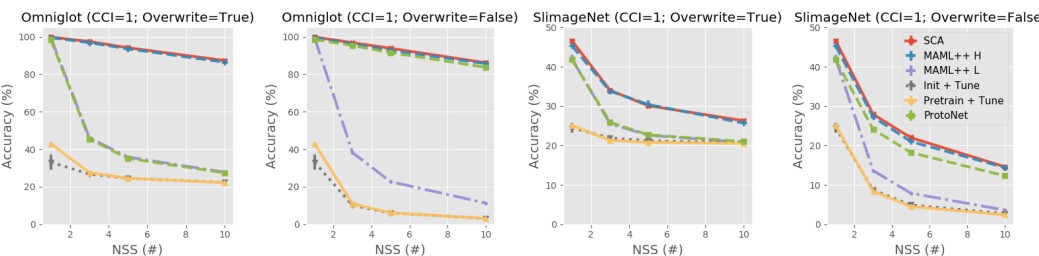

Figure 3: Accuracy (percentage) on the Omniglot and SlimageNet test set for different values of Number of Support Sets Per Task (NSS), Class Change Interval (CCI) equal 1, and with/without overwrite.

even in New Classes with Overwrite (C) where it was previously inferior. This might indicate that in SlimageNet retaining information about previously seen tasks is more important than disentangling complicated super-classes. Overall models that use both embedding-based and gradient-based methods, seem to outperform methods that do just one of the two. In New Classes and Samples (D), embedding-based methods outperform gradient-based ones by a significant margin, while hybrid approaches consistently generate the best performing models. Interestingly, in New Samples (A) the embedding-based and gradient-based methods produce very similar results, whereas in Omniglot gradient-based methods dominated.

**Results: Memory (ATM) and Computational Budgets (MAC)** Figure 4 in Appendix C shows the ATM and MAC costs for NSS $\in [1, 640]$. ProtoNets are the most efficient by two orders magnitude. Other methods (e.g. Low-End MAML++) start off cheaper but increase as the number of support sets incraeses, due to accumulation of features for each new learned class. In terms of ATM it is worth noting that methods such as MAML++ H and SCA tend to become incrementally cheaper than MAML++ L as the number of support sets increases, since they optimize a smaller network in their inner loop. Whereas in terms of MACs MAML++ H and SCA are the most expensive by an order of magnitude or more compared to MAML++ L and ProtoNets.

## 5 CONCLUSION

In this paper, we have introduced a novel benchmark for Continual Few-shot Learning and a new variant of ImageNet, called SlimageNet64, that contains all of 1000 ImageNet classes, but only 200 samples from each class, downscaled to 64×64. Furthermore, we have run experiments on the proposed benchmark, utilizing a number of popular few-shot learning models and baselines. We have found that embedding-based models tend to perform better when incoming tasks contain different classes from one another, whereas gradient-based methods tend to perform better when the task-classes form super-classes of randomly combined categories. Methods utilizing both embedding-based and gradient-based methods (i.e. High-End MAML++ and SCA) outperform methods that use either of the two. In conclusion, we are confident that the proposed benchmark and dataset, will help increasing the rate of progress and the understanding of the behavior of systems trained in a continual and data-limited setting.

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

## A   IMPLEMENTATION DETAILS

**Schedule.** For each model, we used the exact configurations specified in their original papers. For each method (apart from ProtoNets) we used five inner-loop update steps. That is, a Conv-4 architecture for ProtoNets, MAML++ Low End, EWC variants as well as init + fine tune and pretrain + fine tune. In all variants except ProtoNets we adapt the full network's weights in the inner loop. In ProtoNets no inner loop optimization is carried out. For MAML++ High End and SCA we use the improved architecture detailed in Antoniou & Storkey (2019), where there is an image embedding based on a densenet that is not optimized in the inner loop, with a small conv network and a single linear layer which are optimized in the inner loop. For each continual learning task type, we ran experiments on each dataset. Each support set contained 1 sample from 5 classes (5-way, 1-shot) while the target sets contained 5 samples from all the classes seen in a given task. We ran experiments using 1, 3, 5 and 10 support sets for each continual task, therefore creating tasks of increasingly long number of sub-tasks. We ran each experiment 3 times, each time with different seeds for the data-provider and the model initializer. All models were trained for 250 epochs, where each epoch consisted of 500 update steps, each one done on a single continual task, using the default configuration of the Adam learning rule, and weight-decay of 1e-5. At the end of each training epoch we validated a given model by applying it on 600 randomly sampled continual tasks, keeping those tasks consistent across all validation phases. Once all epochs have been completed, we built an ensemble of the top five models across all epochs with respect to validation accuracy, and applied that on 600 random tasks sampled from the test set, to compute the final performance metrics.

**Memory Representations.** We have computed the ATM for each model in section **??**, and in this section we will state the exact way each model represents their ATM bank. ProtoNets represents its memory as class embeddings, called prototypes which store a mean of all vectors associated with any given class, whereas init + fine tune, pretrain + fine tune, MAML++ L and EWC variants, the ATM is represented as the inner loop updatable weights which span the entirety of the Conv-4 architecture used for each model. Finally, for MAML++ H and SCA, the ATM is represented as a single convolutional layer and a single linear layer, which compose the small inner loop updatable network used by the models to adapt to new continual tasks.

**Datasets.** For Omniglot, we used the first 1200 classes for the training set, and we split the rest equally to create a validation and test set. For SlimageNet64, we used 700, 100 and 200 classes to build our training, validation and test sets respectively. The SlimageNet64 splits were chosen such that the training set had mostly living organisms, with some additional everyday tools and buildings, while the validation and test sets contained largely inanimate objects. This was done to ensure sufficient domain-shift between the training and evaluation distributions. As a result this enables a more robust generalization measure to be computed.

## B   COMPARISON BETWEEN DATASETS

We identified four desiderata that a dataset should have in order to be appropriate for CFSL. Note that, it is hard to define quantitative criteria. Here, we provide qualitative criteria that should be considered as generic desiderata therefore subject to a certain degree of arbitrariness.

Table 2: Dataset comparisons. Details details: number of classes in the whole dataset (**#Classes**), number of samples per class (**#Samples**), **Resolution**, **Format**, **Size** allocation of RAM for the whole dataset. Suitability: class diversity (**Diversity**), enough classes (**#Classes**), enough samples (**#Samples**), proper size (**Size**). Omniglot and SlimageNet64 are the best choices for the tasks on grayscale and RGB datasets, respecitively.

| Dataset | Dataset details | | | | | Suitability (satisfies criteria) | | | |
|---|---|---|---|---|---|---|---|---|---|
| | #Classes | #Samples | Resolution | Format | Size (GB) | Diversity | #Classes | #Samples | Size |
| MNIST (LeCun, 1998) | 10 | 7000 | 28×28 | Grayscale | ∼0.20 | ✗ | ✗ | ✗ | ✓ |
| Fashion MNIST (Xiao et al., 2017) | 10 | 7000 | 28×28 | Grayscale | ∼0.20 | ✗ | ✗ | ✗ | ✓ |
| **Omniglot** (Lake et al., 2015) | 1622 | 20 | 28×28 | Grayscale | ∼0.095 | ✓ | ✓ | ✓ | ✓ |
| CUB-200 (Welinder et al., 2010) | 200 | 20-39 | ∼475× ∼400 | RGB | ∼13 | ✗ | ✗ | ✗ | ✓ |
| Mini-ImageNet (Vinyals et al., 2016) | 100 | 600 | 84×84 | RGB | ∼4.7 | ✗ | ✗ | ✓ | ✓ |
| Tiered-ImageNet (Ren et al., 2018b) | 608 | 600 | 84×84 | RGB | ∼29 | ✓ | ✓ | ✓ | ✗ |
| CIFAR-100 (Krizhevsky et al., 2009) | 100 | 600 | 32×32 | RGB | ∼0.68 | ✗ | ✗ | ✓ | ✓ |
| CORe50 (Lomonaco & Maltoni, 2017) | 50 | ∼16.5k | 128×128 | RGB-D | ∼30 | ✗ | ✗ | ✗ | ✗ |
| ILSVRC2012 (Russakovsky et al., 2015) | 1000 | 732-1300 | 224×224 | RGB | ∼800 | ✓ | ✓ | ✗ | ✗ |
| **SlimageNet64 (ours)** | 1000 | 200 | 64×64 | RGB | ∼9.1 | ✓ | ✓ | ✓ | ✓ |

1. **Diversity:** very high degree of diversity in terms of classes. This enforces robustness in the learning procedure, since the model has to be able to deal with previously unseen class semantics. Diversity enable the training, validation, and test splits to lie within different distribution spaces, covering classes that are significantly different from one another.

2. **Number of classes:** high number of categories. This is to ensure that we can train models on CFSL tasks ranging from 1 sub-task, all the way to 100s of sub-tasks. Ideally, the length of a sub-task sequence should not be constrained by the number of classes in the dataset.

3. **Number of samples per class:** fair, but not overabundant, number of samples per class. A dataset with few samples can not capture the difference in distribution within each class (poor evaluation measure), whereas having too many samples per class increases the training time, producing very strong learners but neutralizing the difference among them.

4. **Size:** should be contained. The model should be trained in reasonable time, finances and computational resources. This requirement is crucial to allow use of the dataset by a significant portion of the research community. Here, we define a dataset as appropriate if its size does not exceed 16 GB, which is our reasonable estimate of the average laptop RAM.

## C   MEMORY COST

We report the memory comparison on various datasets in terms of Multiply-Accumulate Computations (MACs) and Across Task Memory (ATM). Measuring MACs is a way to quantify the inference cost of each method. Mesuring ATM is useful for two reasons. (i) We do not restrict an agent to a specific amount of memory, therefore it could easily store all support sets into its memory bank. ATM distinguish models in terms of memory efficiency. (ii) Default measures of computational capacity such (e.g. MACs) are not enough, since they quantify the overall computational requirements and not the actual memory shared across the learning process. This might be minuscule when compared to the model architecture functions which are usually orders of magnitude more expensive. ATM quantifies the efficiency of the learner at compressing incoming data.

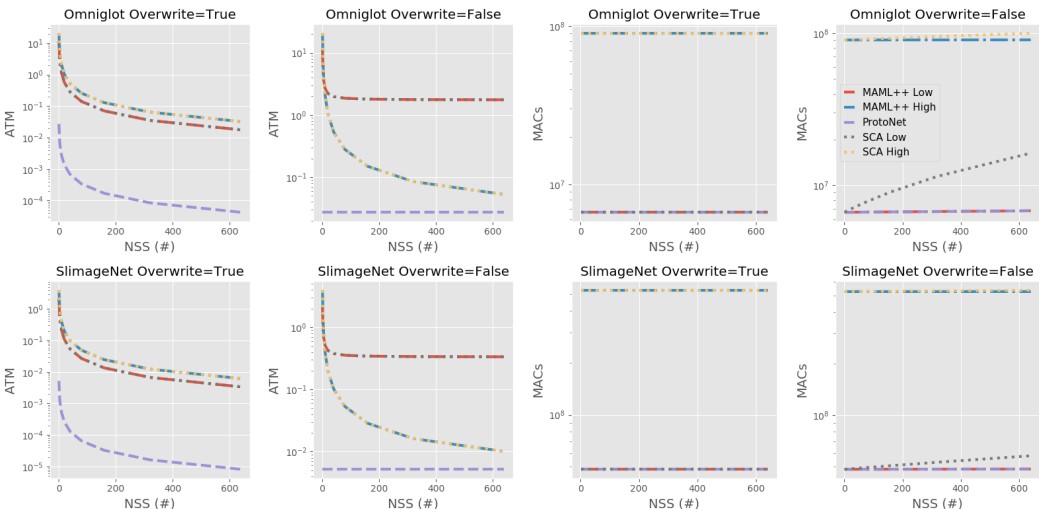

Figure 4: ATM (Across-Task Memory) and MAC (Multiply-Accumulate Computations) costs for a variety of NSS (Number of Support Sets Per Task). ProtoNets are the superior method across the board. In terms of ATM it is worth noting that methods such as MAML++ H and SCA tend to become incrementally cheaper than MAML++ L as the number of support sets increases. Whereas in terms of MACs MAML++ H and SCA are the most expensive by an order of magnitude or more compared to MAML++ L and ProtoNets.

## D   PROPOSED TASKS AND PREVIOUS WORK

We show that our tasks are consistent with the previous work in continual learning (Lomonaco & Maltoni, 2017; van de Ven & Tolias, 2018; Lesort et al., 2019c). Continual learning has

three different scenarios (Lesort et al., 2019c): Single-Incremental-Task (SIT), Multi-Task (MT), Multi Incremental-Task (MIT). The continual learning settings for object recognition fall within the Single-Incremental-Task scenario and can be further partitioned into three update content types (Maltoni & Lomonaco, 2019): New Instances (NI), New Classes (NC), New Instances and New Classes (NIC). In parallel work, van de Ven & Tolias (2018) proposes Non-, Class-, Domain-, and Task- Incremental-Learning (IL) for general continual learning scenarios. This categorization also appears in later work (Hsu et al., 2018; Zeno et al., 2018). We argue that our proposed New-Samples task (A) is most consistent with New Instances (Maltoni & Lomonaco, 2019) but is also similar to Non-IL in (van de Ven & Tolias, 2018) since the distribution of images sampled between support sets does not change between time steps. New-Classes without Overwrite (task B) is most consistent with New-Classes (Maltoni & Lomonaco, 2019) and Class-IL (van de Ven & Tolias, 2018). New-Classes with Overwrite (task C) is not defined in (Maltoni & Lomonaco, 2019) directly but could be seen as a generalization of New Instances where each temporal batch of data (in our case, support sets) is generated from super-classes as opposed to class with a static distribution. New-Classes with Overwrite (task C) is most similar to Domain-IL (van de Ven & Tolias, 2018) where the 'head' dimension (the set of output labels) is kept the same between consecutive batches of data. Finally, our New-Classes with New-Samples task (D) is most similar to New-Classes and Samples (Maltoni & Lomonaco, 2019). This task does not have an equivalent task in (van de Ven & Tolias, 2018). Table 3 shows a disambiguation summary of the continual learning scenarios.

Table 3: Continual Learning task scenarios disambiguation.

| Task | van de Ven & Tolias (2018) | | | Maltoni & Lomonaco (2019) | | |
|---|---|---|---|---|---|---|
| | Class-IL | Domain-IL | Task-IL | SIT-NI | SIT-NC | SIT-NIC |
| NS (A) | | | | ✓ | | |
| NC w/o O (B) | ✓ | | | | ✓ | |
| NC w/ O (C) | | ✓ | | | | |
| NCwNS (D) | | | | | | ✓ |

## E    PSEUDOCODE

---

**Algorithm 1:** Sampling a Continual Few-Shot Task

---

**Data:** Given labeled dataset $\mathcal{D}$, number of support sets per task $NSS$, number of classes per support set $N_C$, number of samples per support set class $K_S$, number of samples per class for target set $K_T$, class change interval $CCI$, and class overwrite parameter $O$

$a = 1, b = 1$;

**for** $a \leq (NSS/CCI)$ **do**

    Sample and remove $N_C$ classes from $\mathcal{D}$;

    **for** $b \leq CCI$ **do**

        $n \leftarrow a \times CCI + b$

        Sample $K_S + K_T$ samples for each of $N_C$ classes

        Build support $\mathcal{S}_n$ with $K_S$ samples per class;

        Build target $\mathcal{T}_n$ with $K_T$ samples per class;

        **if** $O = TRUE$ **then**

            Assign labels $\{1, \ldots, N_C\}$ to the classes;

        **else**

            Assign labels $\{1 + (a - 1) \times N_C, \ldots, N_C \times a\}$ to the classes;

        **end**

        Store sets $\mathcal{S}_n$ and $\mathcal{T}_n$;

    **end**

**end**

Combine all target sets $\mathcal{T} = \bigcup_{n=1}^{N_G} \mathcal{T}_n$

Return $(\mathcal{S}_{1 \ldots N_G}, \mathcal{T})$;

---

