# OpenReview forum: "Defining Benchmarks for Continual Few-Shot Learning"
_ICLR.cc/2021/Conference — Reject_

### Official Review · AnonReviewer1 · 2020-10-24

**Rating:** 5
**Confidence:** 4

**Review:**

----------------------------------
**Summary**

This paper proposes a benchmark for a new task called continual few-shot learning. The benchmark is based on the ImageNet dataset. Basically, the model looks at a part of the support set one after another sequentially, and it is then evaluated on the query set that contains balanced samples from each part of the support set. Under the benchmark, there are four types of challenges, which differs in how they sub-partition the support set. A suite of models have been run and evaluated, including MAML, ProtoNet and SCA. SCA is found to have the best performance, whereas ProtoNet is found to be the most resource efficient.

----------------------------------
**Strengths**

1. I agree that continual few-shot learning is a very useful way of framing few-shot learning problems, especially in the domain of general purpose robotics.

2. Well defined metrics that consider both accuracy and computation resources.

3. There is a significant amount of effort in terms of defining four different scenarios and coming up with another split of the ImageNet dataset and running a number of models on the benchmark.

----------------------------------
**Weaknesses**

1. Limitations in the setup.

    a) Although the paper motivates the application of continual few-shot learning, the proposed benchmark seems rather far away from application. The two applications mentioned in the paper are 1) online shopping for user preference and 2) human-robot interaction for task learning. Why not directly target these settings?

    b) Although the paper is named for defining the benchmark for continual few-shot learning, it is mainly targeting object classification, but not learning new tasks (which is motivated in the human-robot interaction).

    c) The setups in Figure 2 don't seem very natural. Why wouldn’t the model see any old classes in B? Why would the sequence completely overwrite all the class definitions in C? C doesn’t fully capture another very interesting yet unexplored area of domain adaptation, e.g. change the image style from real to synthetic images over time. I can see it is partially addressed in OSAKA between meta-training and meta-test but not in the sequence.

    d) Why is the query set only happening at the very end but not in the middle of the sequence?

2. A lack of new components in the benchmark. The benchmark seems very similar to previous benchmarks (e.g. class incremental learning and CORe50) with the main difference being low-data. According to Table 3, the only setting missing in CORe50 is setting C, which is a rather limited setting since it is extremely unnatural that we need to completely overwrite a label for another class (see my first comment). So what prevents people from using less number of images per class in CORe50, with a possibly pretrained representation from somewhere else, if they want to study a low-shot setting? Pretrained representation from supervised classification seems like a pretty robust approach for few-shot learning based on recent studies (Chen et al., 2019).

3. From an implementation point of view, the proposed benchmark seems no different from a few newly defined mini-batch samplers for regular few-shot learning episodes. If it is no different from a few newly defined mini-batch samplers, then what prevents a model from storing all the examples in the episode? Since this is low-data, it doesn’t seem to have a huge memory cost. Let’s say for each class we store a fixed constant K exemplars, then the overall memory storage is still on the same order as ProtoNet, which stores a single mean vector per class. At the very least, the author should provide results for such a model, as an offline oracle.

4. A lack of models evaluated.

    a) If it can potentially include models that store some of the examples, then there seems no reason to exclude other baselines such as MatchingNet (Vinyals et al., 2016), infinite mixture prototypes (Allen et al., 2019), which can probably address the challenges that ProtoNet faces when dealing with type C sequences.

    b) There have also been a bunch of incremental class learning baselines that can be covered (which doesn’t require meta-learning). For example, iCaRL (Rebuffi et al., 2017), LwF (Li & Hoiem, 2018), BiC (Wu et al., 2019), PODNet (Douillard et al., 2020). Currently I only see EWC as a pretty weak baseline in continual learning and it wasn’t entirely designed for class incremental learning.

    c) I do notice that there are Pre. & Tune and EWC-Pre. baselines that use pretrained representation. This is good. However, it doesn’t explain whether their failures are due to the sequential gradient descent steps or the lack of generalizability of the representation. I suspect it is probably the first case, and have the authors considered using pretrained representation and then using ProtoNet or some other FSL methods at test time?

----------------------------------
**Minor comments**

1. 64x64 seems rather small resolution. I would suggest at least 84x84 for ImageNet images.
2. The memory utilization results should go into the main paper, along with other baselines I suggested that use some example-based storage.
3. Section 2.3, Page 4, Line 13: thie line -> this line.

----------------------------------
**Conclusion**

In conclusion, based on the weaknesses I mentioned above, my score is 5. The paper is one step towards a more useful type of few-shot learning, and I am glad to see the field progresses towards it and I appreciate the effort that the authors have contributed. However, the way it samples new classes and the datasets that the paper study are still in a very limited sense, which prevent the paper from getting direct applications. On the other hand, it is very similar to some other incremental class learning benchmarks and it is not clear what different kinds of conclusions we can draw (e.g. different models that may work better) by using this benchmark instead of using previous ones. Therefore I am leaning towards rejection.

----------------------------------
**References**

- Li, Zhizhong and Hoiem, Derek. Learning without forgetting. In ECCV 2016.
- Vinyals, Oriol, Blundell, Charles, Lillicrap, timothy, Kavukcuoglu, Koray and Wierstra, Daan. Matching networks for one shot learning. In NIPS 2016.
- Rebuffi, Sylvestre-Alvise, Kolesnikov, Alexander, Sperlm, Georg and Lampert, Christoph H. iCaRL: Incremental classifier and representation learning. In CVPR 2017.
- Chen, Wei-Yu, Liu, Yen-Cheng, Kira, Zsolt, Wang, Yu-Chiang and  Huang, Jia-Bin. A closer look at few-shot classification. In ICLR 2019.
- Wu, Yue, Chen, Yinpeng, Wang, Lijuan, Ye, Yuancheng, Liu, Zicheng, Guo, Yandong and Fu, Yun. Large scale incremental learning. In CVPR 2019.
- Allenn, Kelsey R., Shelhamer, Evan, Shin, Hanul and Tenenbaum, Joshua B. Infinite mixture prototypes for few-shot learning. In ICML 2019.
- Douillard, Arthur, Cord, Matthieu, Ollion, Charles, Robert, Thomas and Valle, Eduardo. PODNet: Pooled outputs distillation for small-tasks incremental learning. In ECCV 2020.

---

> ### Author Response · Authors · 2020-11-24
> **Rebuttal Comment 1**
>
> Thank you for your time and effort in providing this review, especially given the instability of the current world-landscape.
>
> Weaknesses
> Limitations in the setup.
> a) Although the paper motivates the application of continual few-shot learning, the proposed benchmark seems rather far away from application. The two applications mentioned in the paper are 1) online shopping for user preference and 2) human-robot interaction for task learning. Why not directly target these settings?
>
> >>In our opinion a benchmark represents an abstraction with respect to real world problems. We want to stress the fact that each one of the four settings we have described in the paper have been carefully built by considering the literature on few-shot and continual learning. Given the space constraints we could not detail the similarities between our tasks and previous work in the main body of the paper, but we included a detailed analysis in the supplementary material (Appendix D). Our benchmark is unifying two fields, and each task provides a setting for studying specific problems. Additional details about each task are given in the answers below. We are open to improve the clarity of the paper on this point, if the reviewer thinks this is necessary.
>
>
> b) Although the paper is named for defining the benchmark for continual few-shot learning, it is mainly targeting object classification, but not learning new tasks (which is motivated in the human-robot interaction).
>
> >> We used image few-shot learning as the general prototype of tasks and then defined different tasks based around the idea of different classes/samples that were to be learned in sequence. The motivating examples we have given are just possible use-cases of our benchmark; this represents an abstraction and can be easily reframed to tackle specific applications.
>
> c) The setups in Figure 2 don't seem very natural. Why wouldn’t the model see any old classes in B? Why would the sequence completely overwrite all the class definitions in C? C doesn’t fully capture another very interesting yet unexplored area of domain adaptation, e.g. change the image style from real to synthetic images over time. I can see it is partially addressed in OSAKA between meta-training and meta-test but not in the sequence.
>
> >> We have carefully defined the various setups by following the literature on few-shot and continual learning. Regarding setting B, the idea is to have a sort of “implicit context” which can be presented multiple times to the learner. This is rather common in a continual setting, where the learner may be exposed to the same set of classes periodically through time. Setting C has been previously used in the Continual Learning literature (Farquhar & Gal, 2018). This setting corresponds to the shared-head condition mentioned by Farquhar & Gal (2018), which makes the task much more difficult since there is overlapping between classes. The motivation for this setting is to have a case that explicitly accounts for limits in the output space of the classifier, since the total number of classes may be an information that is not available in advance in a continual setting. We are keen to improve these points in the paper. Reference: Farquhar, S., & Gal, Y. (2018). “Towards robust evaluations of continual learning”. arXiv preprint arXiv:1805.09733.
>
>
> d) Why is the query set only happening at the very end but not in the middle of the sequence?
>
> >> We care about the performance of the system on previously unseen instances of all previously learned tasks. One could have multiple query sets after every support set, or at the middle as you suggested, but we decided to use the simplest variant which is the one most people care about. Since one can vary the number of support sets before evaluation, someone that cares about performance after only N tasks can achieve that by choosing a suitable NSS.
>
> A lack of new components in the benchmark. The benchmark seems very similar to previous benchmarks (e.g. class incremental learning and CORe50) with the main difference being low-data. According to Table 3, the only setting missing in CORe50 is setting C, which is a rather limited setting since it is extremely unnatural that we need to completely overwrite a label for another class (see my first comment). So what prevents people from using less number of images per class in CORe50, with a possibly pretrained representation from somewhere else, if they want to study a low-shot setting? Pretrained representation from supervised classification seems like a pretty robust approach for few-shot learning based on recent studies (Chen et al., 2019).

---

> ### Author Response · Authors · 2020-11-24
> **Rebuttal Comment 2**
>
> >> We do not restrict people to not use pretrained representation. Furthermore, setting C represents a situation where the model is learning a class of superclasses. For example, a learner starts learning only cars but then motorcycles and boats are added to the class. The model should be able to learn the general idea of human-made modes of transportation.
> Setting C has been previously used in the Continual Learning literature (Farquhar & Gal, 2018) and represents the rather common case where the number of total classes is not known in advance.
>
> From an implementation point of view, the proposed benchmark seems no different from a few newly defined mini-batch samplers for regular few-shot learning episodes. If it is no different from a few newly defined mini-batch samplers, then what prevents a model from storing all the examples in the episode? Since this is low-data, it doesn’t seem to have a huge memory cost. Let’s say for each class we store a fixed constant K exemplars, then the overall memory storage is still on the same order as ProtoNet, which stores a single mean vector per class. At the very least, the author should provide results for such a model, as an offline oracle.
>
> >>What stops them is the fact that we explicitly state that our benchmark prohibits that.
> Note that, even though the number of datapoints for each task is limited, the cost of storing those samples would grow linearly with the length of the episode becoming soon unmanageable. Note that, this is the first benchmark to carefully consider different types of memory metrics. We have explicitly defined various metrics to account for memory costs in Section 3.3 and we have performed several experiments (including on ProtoNet) which are reported in Appendix C.
>
> A lack of models evaluated.
> a) If it can potentially include models that store some of the examples, then there seems no reason to exclude other baselines such as MatchingNet (Vinyals et al., 2016), infinite mixture prototypes (Allen et al., 2019), which can probably address the challenges that ProtoNet faces when dealing with type C sequences.
> b) There have also been a bunch of incremental class learning baselines that can be covered (which doesn’t require meta-learning). For example, iCaRL (Rebuffi et al., 2017), LwF (Li & Hoiem, 2018), BiC (Wu et al., 2019), PODNet (Douillard et al., 2020). Currently I only see EWC as a pretty weak baseline in continual learning and it wasn’t entirely designed for class incremental learning.
> Previous comment on compute.
> c) I do notice that there are Pre. & Tune and EWC-Pre. baselines that use pretrained representation. This is good. However, it doesn’t explain whether their failures are due to the sequential gradient descent steps or the lack of generalizability of the representation. I suspect it is probably the first case, and have the authors considered using pretrained representation and then using ProtoNet or some other FSL methods at test time?
>
> >>We agree that more baselines would be beneficial, however, due to compute constraints we can only add 1 or at most 2 more continual learning baselines. We are considering OML and iCaRL, but that can’t happen within the review timeframe, but would be done should the paper be accepted.

---

### Official Review · AnonReviewer3 · 2020-10-24
**Review for "Defining Benchmarks for Continual Few-Shot Learning"**

**Rating:** 6
**Confidence:** 4

**Review:**

This paper proposes a new machine learning setting called “Continual Few-Shot Learning” which fuses the up until now disparate paradigms of continual learning and few-shot learning. To evaluate methods in this new setting, a new benchmark and dataset called SlimageNet64 are defined. Various methods are evaluated on the new benchmark establishing a set of baseline results for the new setting.

**Pros:**
- The idea of fusing the previously separate domains of few-shot learning and continual learning in the form of a new benchmark is fantastic.
- The benchmarks include metrics for memory usage and computational complexity which are critical in practice and almost always omitted in other benchmarks.
- The paper is well written and straightforward to understand.

**Concerns:**

(1) The benchmark has significant limitations:
- The benchmark focuses on classification accuracy, memory usage, and computational complexity metrics, but has no metrics for other fundamental continual learning metrics such as positive forward transfer, positive backward transfer, forgetting over time, and intransigence (i.e. the inability of an algorithm to learn new tasks) [3,4]. These metrics would have to be measured within a task over the course of learning. The proposed benchmark only does evaluation at the end of a task, which is a limitation.
- Section 3.1 implies that shot and way are fixed across support sets. This is a limitation as realistic few-shot and continual learning scenarios require varying shot and way, including some of the motivating scenarios in Section 1.1.
- Certain guidance about the benchmark is not mentioned. For example, what is the guidance on pre-training the learner via meta-learning or large-scale supervised classification on large datasets outside the benchmark? For example, [5] demonstrates promising continual learning results on a few-shot variant of split MNIST and split CIFAR using a few-shot learner that has been meta-trained on meta-dataset [2].
- The 64 x 64 pixel image size choice in the SlimageNet64 dataset is short-sighted. I realize that the intention is to keep the dataset size small, but in the era of low-end smartphones producing 5 million or more pixel images and the ready availability of pre-trained networks at 224 x 224 pixels implies that the benchmark will not have longevity. The images should be available at their natural size and the allow the learner to scale the images down if they are operating in a constrained computational environment.

(2) The experiments are limited:
- All the experiments were carried out with one fixed configuration (5-way, 1-shot). It would be more revealing if a greater number of configurations were explored, including support sets with random shot and random way.
- The tasks consisted of a very small number of support sets (NSS between 3 and 10) in the accuracy experiments. However, the memory cost tests used NSS=640 (a more realistic number), why not report accuracy results on those runs?
- It is peculiar to compare the accuracy of various methods that use very different network backbones in the same table without clearly delineating the differences in the table (i.e. with a horizontal line or column indicating the backbone). I realize that the purpose of these experiments is to establish baselines with various types and levels of learners but the fact that SCA is bolded as the best in the accuracy tables implies that there is a direct (and unfair) comparison being made between methods that should be made within several categories (say low, medium, and high tiers). In any case, dividing up the table into categories and adding detail on architectures used for the various methods would make the presentation of the results more transparent and insightful.

(3) Much detail required for reproducibility of the experiments is missing:
- No detail is given on how the various approaches in the experiments utilize memory within a task. Section 3.3 says: “Most learners will be compressing a given support set, but this is not strictly the case.” and “(e.g. embedding vectors in ProtoNets, and inner loop parameters for MAML)”. It would be beneficial to describe precisely how memory is used in each of the methods.
- No details were provided on how the baseline methods were carried out. For example, how was the fine tuning done in terms of network architecture, learning rate, optimization method, number of iterations, any memory that is used, were all weights in the network optimized or just the top layer, etc. The same goes for the other methods.
- How is $|\mathcal{G}^x|$ measured? i.e. Is it measured in its tensor form (i.e. 4-byte per channel float) or in its native form (1 unsigned byte per channel)?

**Minor Comments:**
- In the introduction, you should mention meta-dataset [2] along with the other few-shot learning benchmarks listed as it is arguably the most challenging/meaningful one at the moment.
- Section 2.1 should cite [1] as a thorough few-shot learning survey.
- Section 3.2, the acronym NI is used in the sentence before it is defined.
- In Section 3.3, Multiply-Addition operations(MACs) paragraph, it says “…it measures the memory footprint that the model”. Should it be something like “…computational complexity of the model”?
-  Section 3.3, 1st sentence – should it say: “all the task types of interest.” instead of “all the tasks of interest”?
- Should the title of Figure 1 be “High level overview of a CFSL task” instead of “High level overview of the proposed benchmark”? i.e. ‘benchmark’ is a bit too general as there is no other mention of key parts of the benchmark including memory and computational complexity metrics.

**References:**
[1] Hospedales, Timothy, et al. "Meta-learning in neural networks: A survey." arXiv preprint arXiv:2004.05439 (2020).
[2] Triantafillou, Eleni, et al. "Meta-dataset: A dataset of datasets for learning to learn from few examples." arXiv preprint arXiv:1903.03096 (2019).
[3] Schwarz, Jonathan, et al. "Progress & compress: A scalable framework for continual learning." arXiv preprint arXiv:1805.06370 (2018).
[4] Chaudhry, Arslan, et al. "Riemannian walk for incremental learning: Understanding forgetting and intransigence." Proceedings of the European Conference on Computer Vision (ECCV). 2018.
[5] Requeima, James, et al. "Fast and flexible multi-task classification using conditional neural adaptive processes." Advances in Neural Information Processing Systems. 2019.

---

> ### Author Response · Authors · 2020-11-24
> **Rebuttal Comment 2**
>
> (1) The benchmark has significant limitations:
> The benchmark focuses on classification accuracy, memory usage, and computational complexity metrics, but has no metrics for other fundamental continual learning metrics such as positive forward transfer, positive backward transfer, forgetting over time, and intransigence (i.e. the inability of an algorithm to learn new tasks) [3,4]. These metrics would have to be measured within a task over the course of learning. The proposed benchmark only does evaluation at the end of a task, which is a limitation.
>
> >>We are in no way implying that others cannot discuss and report other evaluation measures and metrics while using the benchmark. However many of the other measures are hypothesised as important as proxies for understanding performance. However there is always some discussion as to whether such additional metrics are good measures. We believe the core evaluation of a benchmark should stick to evaluating the task at hand, while of course allowing any other metrics people may wish to compute.
>
>
> Section 3.1 implies that shot and way are fixed across support sets. This is a limitation as realistic few-shot and continual learning scenarios require varying shot and way, including some of the motivating scenarios in Section 1.1.
>
> >>Another fair point. We wanted our scenarios to be more structured and simple, both for computational feasibility and easier interpretations of the results. Future work can propose new benchmarks that contain a stochastic sampling of shot and way. Note that, those settings could be easily defined starting from our benchmark.
>
> Certain guidance about the benchmark is not mentioned. For example, what is the guidance on pre-training the learner via meta-learning or large-scale supervised classification on large datasets outside the benchmark? For example, [5] demonstrates promising continual learning results on a few-shot variant of split MNIST and split CIFAR using a few-shot learner that has been meta-trained on meta-dataset [2].
>
> >>Researchers are allowed to do whatever they want with their model training scheme as long as they are not training on the test set. If another dataset was used for pre-training, it should be stated in the relevant paper.
>
> The 64 x 64 pixel image size choice in the SlimageNet64 dataset is short-sighted. I realize that the intention is to keep the dataset size small, but in the era of low-end smartphones producing 5 million or more pixel images and the ready availability of pre-trained networks at 224 x 224 pixels implies that the benchmark will not have longevity.
>
> >>Producing the table in our paper required 200+ GPU days with the 64x64 images. We want the benchmark to be applicable to people with just a few GPUs such as PhD students, and not only large corporations. That is why we used the 64x64 size. Someone with more compute is always free to create or use a larger dataset.
>
> (2) The experiments are limited:
> All the experiments were carried out with one fixed configuration (5-way, 1-shot). It would be more revealing if a greater number of configurations were explored, including support sets with random shot and random way.
>
> >> An excellent point and suggestion. We appreciate the excellent suggestion of the reviewer. We will improve the presentations of the results in the tables following the reviewer’s recommendations. The results for the 5-way, 1-shot case have taken 200+ GPU-days to be collected. There is a computational barrier in repeating the experiments for all the same set of conditions and methods.
>
> (3) Much detail required for reproducibility of the experiments is missing:
> No detail is given on how the various approaches in the experiments utilize memory within a task. Section 3.3 says: “Most learners will be compressing a given support set, but this is not strictly the case.” and “(e.g. embedding vectors in ProtoNets, and inner loop parameters for MAML)”. It would be beneficial to describe precisely how memory is used in each of the methods.
>
> >>Those are described in the original papers. We could explicitly state them in our paper but unfortunately, our text space budget would not allow that. Given this emphasis by the reviewer we now include more details in the appendix.
>
> No details were provided on how the baseline methods were carried out. For example, how was the fine tuning done in terms of network architecture, learning rate, optimization method, number of iterations, any memory that is used, were all weights in the network optimized or just the top layer, etc. The same goes for the other methods.
>
> >> We shall add those details in the paper. Due to space constraints we could not include those details in the main body of the paper and we had included some details in Appendix A.
>
> How is |Gx| measured? i.e. Is it measured in its tensor form (i.e. 4-byte per channel float) or in its native form (1 unsigned byte per channel)?
>
> >> It is measured in its tensor form, we now clarity this in the paper.

---

> ### Author Response · Authors · 2020-11-24
> **Rebuttal Comment 1**
>
> Thank you for your time and effort in providing this review, especially given the instability of the current world-landscape.
>
> All of your minor comments will be incorporated into the paper shortly.

---

### Official Review · AnonReviewer2 · 2020-10-28
**Paper Review**

**Rating:** 6
**Confidence:** 4

**Review:**

Summary
--------------------
The authors propose a new evaluation protocol which generalizes continual learning and few-shot learning. By controlling the 1) number of support sets per task and 2) the rate at witch tasks change, the authors can span a wide variety of settings previously explored in the literature.
The authors proceed to evaluate several meta learning algorithms under this new protocol and provide a detailed analysis of the results.


Pros
-------------------
+ The overall evaluation protocol is well explained. Figure 2 is a beautiful summary of it.
+ The authors monitor both memory and computation. To me this is one of the key novelties of the paper : it's often overlooked in the literature, yet is very important as one can solve catastrophic forgetting given enough compute
+ The authors release clear and concise code (and installation instructions). This is a great first step to encourage other practitioners to use this dataset
+ The results give interesting insights into popular few-shot learning methods like MAML and ProtoNet

Cons
-------------------
- Table 1 is very hard to read. Please consider either removing some entries to make it bigger (as some baselines perform very poorly), or using another way of presenting it.
- The evaluation protocol has very little CL baselines, I'm not sure why that is. I don't consider EWC to be a good representative of CL methods, there should at least some replay algorithm. Also please consider adding OML [1] as a baseline, as it seems appropriate to this setting.
- The memory and MAC results are barely mentioned in the text. I understand this is hard due to space constraints, however a whole page is spent motivating it.

Small suggestion : The paper would be stronger if it had a wide range of baselines, making it also an empirical study. You constructed a nice benchmark, please show us how (more) known methods perform in it! I'm willing to increase my score if this request is met.

Edit : After reading the appendix, it is mentioned that each model is trained for 250 epochs of 500 steps, where each step is done on a single continual learning task. Here a CL task is a support set (or multiple support sets) from the same distribution ? I just want to double check that a given sample is **only seen during this single continual learning task **, i.e. once this task is over the sample is discarded. Moreover I want to double check that once a task / class is seen, it is never revisited again, even when changing epochs.  I think this is the case, however my whole understanding of the paper rests on this assumption.


[1] Javed, Khurram, and Martha White. "Meta-learning representations for continual learning." Advances in Neural Information Processing Systems. 2019.

---

> ### Author Response · Authors · 2020-11-24
> **Rebuttal Comment**
>
> Thank you for your time and effort in providing this review, especially given the instability of the current world-landscape.
>
> Cons:
>
> Table 1 is very hard to read. Please consider either removing some entries to make it bigger (as some baselines perform very poorly), or using another way of presenting it.
>
> >>We have spent a lot of time trying to improve this table, and admittedly, it’s not ideal yet. We’ll try to revise, but we are not entirely sure what more can be done. Feel free to suggest specific ideas and we’d be happy to look into them. We have considered breaking it up into two parts, one smaller for the main paper and one larger for the appendix, but that would mean omitting results that some readers would be interested in, from the main text.
>
> The evaluation protocol has very little CL baselines, I'm not sure why that is. I don't consider EWC to be a good representative of CL methods, there should at least some replay algorithm. Also please consider adding OML [1] as a baseline, as it seems appropriate to this setting.
>
> >> We tried to choose a few representative methods from each setting, admittedly leaning more into simpler methods that also required less compute to benchmark. We agree that the addition of 1-2 more continual learning methods would be a good idea for our work. We’ll work on adding the OML algorithm if this work is accepted since we do not have enough time until the review process is over.
>
> The memory and MAC results are barely mentioned in the text. I understand this is hard due to space constraints, however a whole page is spent motivating it.
>
> >>Very good point. We have now added additional discussions on our memory and MAC results. However, as you have recognized, the availability of space is a problem.
>
>
> Edit : After reading the appendix, it is mentioned that each model is trained for 250 epochs of 500 steps, where each step is done on a single continual learning task. Here a CL task is a support set (or multiple support sets) from the same distribution ? I just want to double check that a given sample is **only seen during this single continual learning task **, i.e. once this task is over the sample is discarded. Moreover I want to double check that once a task / class is seen, it is never revisited again, even when changing epochs. I think this is the case, however my whole understanding of the paper rests on this assumption.
>
> >>This is an issue of the approach used to test the models on the benchmark, but not particularly an issue of the benchmark itself which is agnostic of the approach taken. Your understanding is mostly right. During a continual few-shot learning task, a given sample is only seen once and then discarded, as you noted. During meta-training, at each iteration, we randomly sample the classes in each support set in a manner consistent with the task. Then we randomly sample the individual data points (without replacement within the individual meta-training iteration) from the core dataset, conditioned on each class. Likewise the target dataset is always unseen data points.
>
> >>For a subsequent step of meta-training, this process is repeated. It is possible for the same data point to turn up a second time in a future meta-training iteration, though it is not a strong feature of the process - in SlimageNet64, for each class we are sampling from 200 samples, but there 1000 classes. So it is even rare for individual classes to turn up in a sequence, and even rarer for individual data points. We also do not consider this to be a big issue - it is much more important that there is no replacement within a run. Do let us know if there is some concern you have here that you would like us to address.

---

### Official Review · AnonReviewer4 · 2020-10-29
**A benchmak for continual few-shot learning**

**Rating:** 4
**Confidence:** 3

**Review:**

The work proposes Continual Few-Shot learning -- a setting to study tasks (1) with a small labeled dataset, and (2) retain knowledge acquired on a sequence of instances.  Additionally, the authors build a compact variant of ImageNet which retains all original 1000 classes but only contains 200 instances of each one (a total of 200K data-points) downscaled to 64 × 64 pixels.

By evaluating baselines on the proposed benchmark, the authors observe that embedding-based models tend to perform better when "incoming tasks contain different classes from one another" and gradient-based methods tend to perform better when the task classes "form super-classes of randomly combined categories."

The overall idea is interesting (few-shot + sequential observations); however, it's not clear should one take home after reading this draft. The conclusions seem intuitive and reasonable, but leave the reader with questions about the main findings of the work.

I would take issue with the way the word "continual learning" is used. In practice, the author(s) use existing datasets where the instances arrive in a sequential manner (streaming observations). However, this is not quite what they motivate: "Consider a user in a fast-changing environment who must learn from the many scenarios that are encountered." since, in the described sequential setting all the instances belong to the same underlying distribution (the original dataset), even though they're observed sequentially.

Note that a realistic temporal observations are much more challenging (e.g., the language of search queries over time because of the change in the functionality of search engines, or the changing distribution of images over time because of various social changes.)

The overall direction is promising: clearly, we need to move towards more data-efficiency and build better frameworks for measuring the generalization of our models. However, I am not convinced if the presented work is a significant step toward that goal (or, at least, I don't see it). Happy to change my mind, if I am missing anything.

---

> ### Author Response · Authors · 2020-11-24
> **Rebuttal Comment**
>
> Thank you for your time and effort in providing this review, especially given the instability of the current world-landscape.
>
> Reply to Reviewer #4:
>
> The overall idea is interesting (few-shot + sequential observations); however, it's not clear should one take home after reading this draft. The conclusions seem intuitive and reasonable, but leave the reader with questions about the main findings of the work.
>
> >>We thank the reviewer for showing their interest in the overall idea. We are keen to improve the clarity of the paper, we would appreciate it if the reviewer could provide more detailed feedback on this point to facilitate our work.
>
> I would take issue with the way the word "continual learning" is used. In practice, the author(s) use existing datasets where the instances arrive in a sequential manner (streaming observations). However, this is not quite what they motivate: "Consider a user in a fast-changing environment who must learn from the many scenarios that are encountered." since, in the described sequential setting all the instances belong to the same underlying distribution (the original dataset), even though they're observed sequentially.
>
> >>Our benchmark provides a wide range of task types, some of which explicitly ‘overwrite’ classes, therefore ensuring that the distribution changes as learning is happening. Furthermore, even in the non overwriting settings, since the examples are observed in a single-shot manner, each observed support set will usually contain the same class in a very different situation, i.e. dog in park vs dog on the pier vs dog on the floor at a house. Furthermore, it’s worth stressing that our benchmarks remain a bit more abstract, and general in order to establish generalization metrics that can be used to judge the performance of the system in a wide variety of contexts. Then, once a model is judged on a general benchmark, it can be used in a specific application setting to establish a more specific, application-directed generalization metric. In a sense, we want benchmarks to be general.

---

### Decision · Program_Chairs · 2021-01-07
**Final Decision**

**Decision:**

Reject

**Comment:**

The reviewers appreciate the steps taken to combine continual learning with few-shot learning, this is an interesting intersection with many potential applications. However, the reviewers generally outlined a number of concerns with the benchmark and paper in its current form. They largely feel that this benchmark doesn’t differentiate itself well enough from other incremental learning benchmarks, nor does it experiment with a wide enough variety of settings (additional episode configurations, more FSL/CL approaches). As such, it is difficult to determine at this point what major insights can be gained from this benchmark. I understand that there is a tradeoff: computation is limited, and there is merit in keeping things simple. However, the general consensus is that more work needs to be done in order to fully realize the potential of this benchmark.